# The global Deep-time Sediment Nitrogen Isotopes in Marine Systems (DSMS-NI) database

Yong Du[1], Huyue Song[1,*], Thomas J. Algeo[1,2,3], Hui Zhang[1], Jianwei Peng[1], Yuyang Wu[1,4], Jiankang Lai[1], Xiang Shu[1], Hanchen Song[1], Lai Wei[5], Jincheng Zhang[6], Eva E. Stüeken[7], Stephen E. Grasby[8], Jacopo Dal Corso[1], Xiaokang Liu[1], Daoliang Chu[1], Li Tian[1], Qingzhong Liang[6], Xinchuan Li[6], Hong Yao[6], Haijun Song[1]

[1]State Key Laboratory of Geomicrobiology and Environmental Changes, School of Earth Sciences, China University of Geosciences, Wuhan 430074, China

[2]Department of Geosciences, University of Cincinnati, Cincinnati, OH 45221-0013, USA

[3]State Key Laboratory of Oil and Gas Reservoir Geology and Exploitation, Chengdu University of Technology, Chengdu 610059, China

[4]College of Marine Science and Technology, China University of Geosciences, Wuhan 430074, China

[5]Shool of Future Technology, China University of Geosciences, Wuhan 430074, China

[6]School of Computer Science, China University of Geosciences, Wuhan 430074, China

[7]School of Earth & Environmental Sciences, University of St Andrews, St Andrews KY16 9AL, UK

[8]Geological Survey of Canada, Natural Resources Canada, Calgary, Alberta T2L 2A7, Canada

*Correspondence to:* Huyue Song (hysong@cug.edu.cn)

**Abstract.** Stable nitrogen isotope records preserved in marine sediments provide critical insights into Earth's climate history and biospheric evolution. Although numerous studies have documented nitrogen isotope ($\delta^{15}N$) records for various geological systems (Archean to Recent) and paleogeographic settings, the scientific community remains constrained by the absence of a standardized database to systematically investigate their spatiotemporal evolution. Here, we present the database of Deep-time Sediment Nitrogen Isotopes in Marine Systems (DSMS-NI), a comprehensive global compilation of $\delta^{15}N$ data and associated geochemical parameters, spanning a vast collection of sediment samples dating from the Recent to the Archean. This database encompasses 70 854 $\delta^{15}N$ records derived from 417 publications, systematically organized with 31 metadata fields categories (e.g., chronostratigraphic ages, coordinates, lithology, metamorphic grade, sedimentary facies, references) encompassing 1 999 226 metadata. This repository further incorporates 130 proxy data fields, including 281 215 geochemical data spanning total organic carbon (TOC), total nitrogen (TN), and organic carbon isotopes ($\delta^{13}C_{org}$), major and trace elements and iron species. These integrated parameters enable

evaluation of sample fidelity and factors influencing $\delta^{15}N$ signatures. The DSMS-NI database will facilitate research for key geological intervals such as the Permian/Triassic boundary and the Cretaceous oceanic anoxic events (OAEs). Researchers can leverage temporal and paleogeographic information, alongside geochemical data, to conduct spatiotemporal analyses, thereby uncovering changes in deep-time marine nitrogen cycles and paleoenvironmental conditions. The database is open-access via the Geobiology portal (https://geobiologydata.cug.edu.cn/, last access: 30 April 2025), allowing users to access data and submit new entries to ensure continuous updates and expansion. This resource represents a vital foundation for studies in paleoclimate, paleoenvironment, and geochemistry, offering essential data for understanding long-term Earth-system processes. The data files described in this paper are available at https://doi.org/10.5281/zenodo.15117375 (Du et al., 2025a).

## 1 Introduction

Nitrogen, as an essential nutrient and redox-sensitive element, plays a crucial role in biological evolution and environmental climate changes (Ader et al., 2016; Pellerin et al., 2024). Typically, nitrogen isotope compositions are reported as a relative deviation of sample's isotopic ratio relative to that of atmospheric $N_2$, expressed in per mille (‰) as $\delta^{15}N = (R_{sample}/R_{AIR-N2} - 1) \times 1000$ ‰, where R = $^{15}N/^{14}N$. The $\delta^{15}N$ record has become one of the primary tools for tracing the evolution of the nitrogen cycle and reconstructing redox conditions through deep time (Algeo et al., 2014; Sahoo et al., 2023; Du et al., 2024; Moretti et al., 2024). Advances in analytical techniques have facilitated rapid growth in the application of $\delta^{15}N$ for paleoenvironmental studies in recent decades (Fig. 1; Zhong et al., 2023). Given nitrogen's short marine residence time of approximately 3000 years, which leads to regionally variable and rapidly shifting patterns (Gruber and Galloway, 2008), high-resolution $\delta^{15}N$ datasets with detailed temporal and spatial coverage are critical for elucidating nitrogen cycle dynamics through Earth history.

Existing compilations of deep-time marine $\delta^{15}N$ records exhibit significant limitations in term of temporal coverage and metadata compliance. Previous efforts have focused specifically on Precambrian to investigate the origins of microbial nitrogen metabolism and redox evolution during the Great Oxygenation Event (Thomazo et al., 2011; Stüeken et al., 2016, 2024; Kipp et al., 2018; Uveges et al., 2025). Other studies have targeted Phanerozoic systems (Algeo et al., 2014) or specific intervals such as the Paleozoic (Koehler et al., 2019), Cambrian (Wang et al., 2018; Liu et al., 2020), Carboniferous (Algeo et al., 2008), Triassic (Sun et al., 2024), and Cenozoic (Tesdal et al., 2013) to

analyze key biological and environmental events. The largest compilation of data from pre-Cenozoic records contains fewer than 8000 $\delta^{15}N$ entries (Stüeken et al., 2024). In contrast, Tesdal et al. (2013) compiled up to 33 352 entries, but all of these records are from the past 6 million years. Moreover, these repositories often fail to adhere systematically to the FAIR (Findable, Accessible, Interoperable, Reusable) data principles (Wilkinson et al., 2016) and offer limited metadata categories. Typically, they provide only broad geologic ages, lithology, and metamorphic grades, while lacking essential metadata such as paleogeographic coordinates, depositional environments, and high-resolution chronostratigraphy (Table 1). Current metadata-rich databases that follow FAIR principles remain limited to fewer than 3000 $\delta^{15}N$ entries (e.g., Farrell et al., 2021; Lai et al., 2025), highlighting the urgent need for a rigorously standardized, spatiotemporally comprehensive $\delta^{15}N$ database.

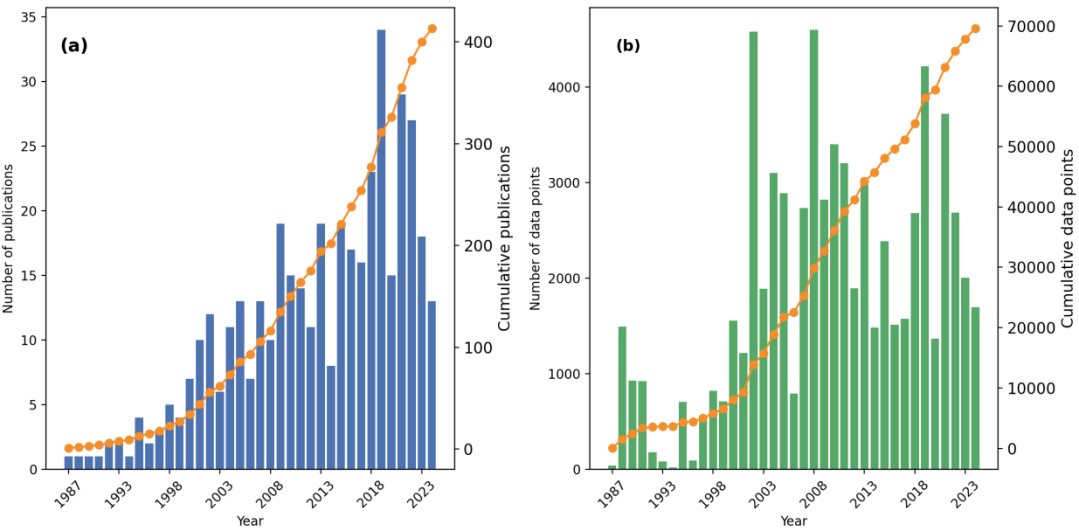

**Figure 1.** Temporal trends in (a) nitrogen isotope publications and (b) $\delta^{15}N$ data entries in the DSMS-NI database. Vertical bars denote annual publication/dataset counts, while dots connected by lines represent cumulative totals over the years.

The DSMS-NI database, a repository of deep-time sediment nitrogen isotopes in marine systems spanning Earth history, aims to address this need. The DSMS-NI database is a part of the broader GBDB (Geobiology Database) project, which aims to build a comprehensive database of biotic and biogeochemical evolution throughout time and to explore the mechanisms driving these evolutionary processes. By integrating detailed metadata, DSMS-NI provides a valuable resource for studying nitrogen cycle evolution and paleoenvironmental conditions at a range of temporal and spatial scales. This compilation provides an extensive survey of $\delta^{15}N$ records on bulk sediments and specific phases in sediments deposited within marine environments, with a particular emphasis on data predating the

Cenozoic Era. Derived from 417 peer-reviewed publications and publicly available datasets, it currently encompasses 70 854 discrete $\delta^{15}N$ measurements for various components (e.g., bulk rock, shell-bound, kerogen). In addition, it includes roughly 281 215 associated data points for carbon, sulfur isotopes, and major and trace element concentrations reported alongside the $\delta^{15}N$ values. Each entry is linked to a comprehensive set of standardized metadata, ensuring consistency and facilitating robust data analyses. Our goal is to make DSMS-NI a dynamic, evolving database that improves over time, with data visualizations updated concurrently on the Geobiology Data website (https://geobiologydata.cug.edu.cn/, last access: 30 April 2025).

**Table 1** Overview of deep-time $\delta^{15}N$ complication.

| Data Source | Number of $\delta^{15}N$ records | Metadata | Spatial range | Temporal range |
|---|---|---|---|---|
| Tesdal et al. (2013) | 33 352 | Fine age; Modern coordinate; Site | Global | Neogene to Present |
| Algeo et al. (2014), restricted access | 6006 | Broad age; Formation | Global | Ediacaran to Present |
| Stüeken et al. (2016) | 6449 | Broad age; Formation; Lithology; Metamorphic grade | Global | Since the Paleoarchean |
| Stüeken et al. (2024) | 10 584 | Broad age; Formation; Lithology; Metamorphic grade | Global | Since the Eoarchean |
| Kipp et al. (2018) | 6468 | Broad age; Formation; Lithology; Metamorphic grade | Global | Since the Paleoarchean |
| Koehler et al. (2019) | 2454 | Broad age; Formation; Lithology; Metamorphic grade | Global | Paleozoic |
| Farrell et al. (2021), SGP database | 840 | Broad age; Modern coordinate | Global | Paleozoic and Ediacaran |
| Lai et al. (2025), DM-SED database | 2561 | Fine age; Modern coordinate; Paleocoordinate; Site; Formation; Depositional environments; Lithology; Metamorphic grade | Global | Since the Neoproterozoic |

*Note.* The classification of age resolution in the metadata is as follows:

- Broad age: Age estimates assigned uniformly to data from multiple stratigraphic levels within the

same geological formation, indicating no resolved internal chronological order.
- Fine age: Sequentially ordered ages calculated for individual samples, derived from an established
age-depth model.

Version 0.0.1 of the DSMS-NI database is available in CSV format on Zenodo
(https://doi.org/10.5281/zenodo.15117375), and dynamic updates will be maintained on the
GeoBiology website. The following sections provide a comprehensive overview of the database
compilation methods, data structure, and details of the dataset, including data sources, selection criteria,
and definitions of metadata fields. Additionally, we analyze the temporal and spatial trends of $\delta^{15}N$
within the dataset, discuss potential applications and limitations, and outline the foundation for the
database's continuous development and scientific utility.
**2 Compilation methods**
**2.1 Data compilation**
An extensive search was conducted based on published articles, reports, theses, and datasets to gather
all available literature on deep-time nitrogen isotopes. Initially, a keyword-based search combining
geological period and nitrogen isotope was performed on Google Scholar, yielding over 3000 relevant
literature sources after removing duplicates. A significant portion of the articles, however, only
discussed previously published $\delta^{15}N$ data, rather than presenting newly measured data, which were
manually excluded from the data compilation. Additionally, geochemical databases such as PANGAEA
(https://www.pangaea.de/, last access: April 1 2025), EarthChem (https://www.earthchem.org/, last
access: April 1 2025), SGP (https://sgp-search.io/, last access: April 1 2025), and NOAA
(https://www.ncei.noaa.gov/, last access: April 1 2025) were queried to ensure comprehensive coverage
of dataset sources (Diepenbroek et al., 2002; Gard et al., 2019; Farrell et al., 2021). Where overlaps
existed between datasets and publications, journal articles were prioritized as the primary data sources.
Further filtering excluded studies on non-marine sediments, entries lacking essential metadata (e.g.,
geological age, latitude and longitude), and a limited number of Cenozoic records with inaccessible
data. Ultimately, the curated dataset includes 424 valid sources published between 1983 and 2024,
representing a comprehensive compilation of nitrogen isotope records for deep-time marine sediments.
Data from each publication were stored in various formats, including tables within the main text,
supplementary files, or shared databases. Data extracted from tables and supplementary files were
initially processed by computer algorithms, followed by manual verification and supplementation. For
databases, data files were downloaded manually. In cases where publications did not provide direct
data, data points were extracted from figures using GetData Graph Digitizer (ver. 2.24), and these
entries were labeled as "plot" in the Notes section. Each publication was then organized into an
individual data file with clear labeling of sources and unique site identifiers. These files were
subsequently merged into a master dataset based on standardized column headers. In the final master
dataset, additional metadata were curated, including geological age, latitude and longitude, lithology,
depositional facies, and metamorphic grade. High-resolution ages and paleocoordinates were calculated
and converted, where applicable.
**2.2 Data selection and quality control**
Given that biogeochemical and paleoenvironmental studies based on nitrogen isotopes require the
assessment of the depositional environment and post-depositional alteration, geochemical data apart
from $\delta^{15}N$ are crucial (Tribovillard et al., 2006; Robinson et al., 2012). Therefore, we collected other
contemporaneously published geochemical data of the same samples as $\delta^{15}N$ from the literature
relevant to the formations in our database. All available data from each research site were included as
comprehensively as possible, rather than excluding entries solely due to the absence of $\delta^{15}N$ values.
This approach allows for the potential interpolation of the time-series data. However, geochemistry
fields with fewer than 100 data points in the final compilation were excluded due to their limited
analytical utility, such as Mo and Fe isotopes.
To ensure the reliability and applicability of the data, each entry underwent a rigorous screening
and evaluation process. Initially, we assessed the data source and its spatiotemporal context. All studies
included in the database were required to report verified $\delta^{15}N$ values with clear data provenance and
well-defined spatiotemporal information. Data entries lacking traceable sources were excluded.
Similarly, entries without precise geographic or temporal information were not considered. Data from
geological settings representing highly localized environments or with high metamorphic grades, such
as samples affected by hydrothermal activity (Martin and Stüeken, 2024) or highly metamorphosed
minerals (e.g., mica; Jia and Kerrich, 2000; Busigny et al., 2003), were also excluded given that their
$\delta^{15}N$ compositions likely record alteration processes rather than seawater signatures. This filtering
criterion was applied based on descriptions in the original literature rather than a fixed metamorphic
grade threshold. The $\delta^{15}N$ values for bulk rock and decarbonated rock were classified as primary
entries ($\delta^{15}N_{bulk}$), while values for specific phases, such as fossil shells, kerogen, clay-bound nitrogen,

and porphyrins, were categorized solely as secondary entries ($\delta^{15}N_{sp}$). Only primary entries were analyzed in the data visualizations presented later in this study.

Only $\delta^{15}N$ obtained through standardized, widely accepted techniques were included in the database. These primarily consist of elemental analyzer-isotope ratio mass spectrometry methods applied to bulk rock, decarbonated fractions, or kerogen (Song et al., 2023), as well as denitrifier-based mass spectrometry methods for microfossils (Ren et al., 2012; Smart et al., 2018). Studies employing non-standard or unvalidated methods, such as stepwise combustion (Ishida et al., 2017), were excluded. Data from highly metamorphosed settings (e.g., hydrothermal alteration), terrestrial lakes and rivers, modern organisms and their metabolic products, and liquid phases were flagged and omitted from the database (e.g., Bebout et al., 1999; Chase et al., 2019; Xia et al., 2022). For data from the same site but at different depths or lithologies, or for measurements of different components in the same layer (e.g., bulk sediment and decarbonated sediment), or replicate analyses of the same homogenized sample, each entry was recorded separately to accurately capture variability.

Metadata on paleocoordinates, depositional setting, lithology, and metamorphic grade are included, wherever available. Entries were not excluded due to missing such metadata, as these can potentially be supplemented in future research. When such metadata were not directly reported in the literature, we attempted to estimate them using supplementary data or external sources, such as paleogeographic reconstructions. For entries for which metadata could not be determined, blank values were assigned.

**3 Data summary**

Since nitrogen isotope studies in sediments began in the late 1980s, the number of published studies has shown an accelerating growth trend, doubling approximately every decade. This trend is mirrored by a steady increase in data volume, with an average annual addition of around 2720 data points over the past two decades (Fig. 1). However, the rate of data growth slightly lags behind that of publications, largely because early Ocean Drilling Program (ODP) and Integrated Ocean Drilling Program (IODP) projects contributed substantial datasets within individual publications (e.g., Liu et al., 2008). Ocean drilling remains a vital component of the database, covering geological intervals since the Cretaceous. Some early drilling data were not initially publicly accessible and have been supplemented through existing literature compilations, particularly the substantial dataset from Tesdal et al. (2013), along with enriched metadata.

The DSMS-NI database comprises a total of 31 metadata fields and 130 proxy data fields,
organized into five primary categories (Table 2): (1) sampling location, (2) age information, (3)
geochemical data, (4) lithological characteristics, (5) analytical methods and (6) references. For clarity
and consistency throughout this data descriptor, the term "entries" refers to individual proxy values and
their associated metadata (i.e., rows), while "fields" denote the metadata attributes recorded for each
entry (i.e., columns).

**Table 2** Field names and descriptions.

| Field name | Description |
| --- | --- |
| Sample ID and location fields | |
| SampleID | Unique sample identification code, as originally published |
| SiteName | Name of the drill core site or section |
| SampleName | Author denoted title for the sample (often non-unique, e.g., numbered) |
| Location1 | Detailed location of the data collection site |
| Location2 | Country or ocean of the data collection site |
| Latitude | Modern latitude of collection site rounded to two decimals; negative values indicate the Southern Hemisphere (decimal degrees) |
| Longitude | Modern longitude of the collection site rounded to two decimals; negative values indicate the Western Hemisphere (decimal degrees) |
| Paleolatitude | Paleolatitude of collection site rounded to two decimals; negative values indicate the Southern Hemisphere (decimal degrees) |
| Paleolongitude | Paleolongitude of the collection site rounded to two decimals; negative values indicate the Western Hemisphere (decimal degrees) |
| Age fields | |
| Era | The geologic era, in reference to GTS v202309 |
| Period | The geologic period, in reference to GTS v202309 |
| Epoch | The geologic epoch, in reference to GTS v202309 |
| Stage | The geologic stage, in reference to GTS v202309 |
| Age | Age, in reference to GTS v202309 |
| Formation | Geologic formation name |
| Unit | Specific geologic event layers |

| | |
|---|---|
| RelativeDepth | Stratigraphic height or depth (m) |
| Petrological characteristic fields | |
| Lithology | Lithological name of the sample, as originally published |
| LithType | Lithology type of sample (e.g., carbonate, siliciclastic) |
| MetamorphicGrade | The degree to which the rock has undergone transformation due to heat and pressure conditions |
| Setting | Depositional environment (e.g., epeiric, bathyal) |
| WaterDepth | Estimated depositional water depth of the data collection site |
| Method fields | |
| Material | Samples subjected to $\delta^{15}N$ analysis (e.g., decarbonated sediment, diatom, kerogen) |
| Technique$\delta$15N | Methodology employed for $\delta^{15}N$ measurement (e.g., EA combustion, denitrifier method) |
| Data fields | |
| Isotopes | The isotopic composition expressed in per mille (‰) as $\delta$ (e.g., $\delta^{15}N$, $\delta^{13}C$) |
| Elements | The concentration of elements within rocks (e.g., TN, P, Fe, Cu, Ce) |
| RockEval | Proxies of hydrocarbon potential measured by pyrolysis method (e.g., S1, OI, $T_{max}$) |
| FeSpecies | Concentrations and ratios of different iron species in rocks (e.g., $Fe_{py}$, $Fe_{HR}/Fe_{T}$) |
| Reference fields | |
| FirstAuthor | The last name of the first author of the original publication |
| Year | The year of the original publication |
| Title | The title of the original publication |
| Reference | The formatted reference of the original publication |
| DOI | The DOI of the original publication |
| DataSource | The repository hosting the data except for the original publication |


**Sample ID and Location fields.** Each data entry was assigned a unique Sample ID to distinguish
it from other data entries. Geographic location information includes the modern latitude and longitude
(Latitude and Longitude) referencing WGS84 (World Geodetic System 1984), obtained directly from
original literature or external sources whenever possible. For studies that do not provide exact
coordinates, approximate locations are estimated based on geographic descriptions or accompanying
maps, using tools such as Google Maps. Additionally, we record the broader sampling region (e.g.,
country or oceanic region) and specific sampling site details (such as province, county, or uplift names).
The location fields also include the name of the drilling site or outcrop section (SiteName), which
identifies the precise drilling location or outcrop at which samples were collected, providing valuable
geographic context. Certain SiteNames are uniquely associated with major drilling projects (e.g., ODP,
IODP), which is important for subsequent data supplementation and analysis. Some samples also have
a SampleName, as designated by the original authors—typically a code or non-unique label reflecting
the naming format in the primary literature. Although multiple samples in the database may share the
same SampleName, each entry has a distinct Sample ID to ensure the uniqueness of each record.
We also provide paleolatitude and paleolongitude (PaleoLatitude and PaleoLongitude), calculated
based on the geological age of each sample and using paleogeographic reconstruction tools such as
PointTracker v7.0, built on the plate rotation model of Scotese and Wright (2018). Paleocoordinate data
are crucial for understanding the historical shifts in sample locations and their relationship to
depositional environments (Percival et al., 2022; Li et al., 2025). To maintain consistency, all
geographic coordinates are standardized to two decimal places.
**Age fields.** Each entry includes not only absolute age data but also a series of geologic age-related
fields to provide precise temporal context. These fields enable targeted data retrieval at a range of
geological timescales, facilitating comparisons with newly added data. The GeologicalAge field
captures broad temporal frameworks, recorded as Epoch for the Phanerozoic (e.g., Early Triassic) and
Era for the Precambrian (e.g., Neoproterozoic). For more refined stratigraphic resolution, the Stage
field (e.g, Induan) is used, with the System as a substitute for Precambrian samples (e.g., Ediacaran).
The Age field records the absolute age of each sample, following the International Chronostratigraphic
Chart, GTS v202309. The Formation field notes the geological unit (formation or member) from which
the sample was collected, aiding in understanding its depositional context and relation to surrounding
strata (Murphy and Salvador, 1999). However, Formation data are generally limited to outcrop sections,
as ocean drilling samples lack specific formation designations. The Unit field identifies particular
stratigraphic units or geologic event layers, such as the Cretaceous pre-OAE2 or OAE2 (Jenkyns,
2010), which aids in correlating samples within recognized geological events. The RelativeDepth field
records the sample's relative depth in the section or drill core, which is essential for high-resolution age
analyses and sedimentation rate calculations.

231       Age data allocation follows these guidelines below. When precise ages and geological age

information for each sample were provided in the original source, these values are prioritized. However,
for data from the Common Era (i.e., negative ages), they are uniformly assigned a value of 0 Ma,
meaning that all such data are treated as reference values for modern top sediments. Otherwise, age
assignments follow two methods based on data availability. (1) For records with at least two samples or
stratigraphic horizons of known age (e.g., radiometrically dated layers or well-defined stage
boundaries), we constructed an age-depth model. This model linearly interpolates ages between these
tie-points along the RelativeDepth axis, assuming a constant sedimentation rate within each interval
between stratigraphic age tie-points. While this assumption is effective for establishing the relative
temporal sequence of samples, which is critical for capturing first-order stratigraphic trends, it
necessarily introduces uncertainties in absolute age determination due to potential variability in
sedimentation rates or local stratigraphic features. (2) For records lacking sufficient data for an
age-depth model, a single age was assigned to all samples. When only one age constraint (e.g., a
radiometric date from a nearby stratum) is available, that specific age is applied. In the absence of any
direct age control, the median age of the corresponding geologic stage is used as a default. It should be
noted that assigning a uniform age to a suite of samples, particularly using the median stage age, carries
significant uncertainty, theoretically on the order of the duration of the entire geologic interval (which
can approach 100 Myr for long stages of the Precambrian). Profiles constrained by a single radiometric
date, which is the predominant method for dating sequences older than 600 Ma, are generally more
reliable than those relying solely on a median stage age.

251       **Data fields.** The dataset includes analyses of isotopic compositions, elemental concentrations, and

specific components. To maintain consistency, all units were standardized during data collection, as
original publications sometimes report these data in varying units. (1) Isotopic data include $\delta^{15}N$, $\delta^{13}C$,
$\delta^{18}O$, and $\delta^{34}S$, all expressed in ‰ relative to international standards. Nitrogen isotopes are reported
relative to atmospheric nitrogen (Air $N_2$), carbon and oxygen isotopes relative to the Vienna Pee Dee
Belemnite (VPDB) standard, and sulfur isotopes relative to the Vienna Canyon Diablo Troilite (VCDT)
standard (Hoefs, 2009). (2) Elemental concentrations include TN (total nitrogen), TOC (total organic
carbon), TS (total sulfur), $CaCO_3$, TC (total carbon), TIC (total inorganic carbon), P, Al, K, Si, Ca, Ti,
Na, Mg, Fe, as well as iron species data and LOI (loss-on-ignition), and they are reported in weight
percent (wt %). Concentrations of other trace elements are standardized to parts per million (ppm). (3)
Some data originally reported as oxide concentrations were converted to elemental concentrations
based on stoichiometric ratios, such as $P_2O_5$. (4) Additional derived values include ratios of iron species,
dry bulk density, and rock eval indices (Peters et al., 1986; Poulton and Canfield, 2005). These indices
comprise alkenone content (C37, in nmol/g), oxygen index (OI, mg CO2/g TOC), hydrogen index (HI,
mg HC/g TOC), maximum pyrolysis temperature (Tmax, °C), free hydrocarbons (S1, mg HC/g Rock),
hydrocarbons generated from rock pyrolysis (S2, mg HC/g Rock), and $CO_2$ released from organic
matter pyrolysis (S3, mg CO2/g Rock). Some inaccessible data points were visually extracted from
figures using scatterplot recognition techniques, which are marked as "plot" in the Notes field. Data
with values exceeding detection limits or those erroneous (e.g., negative values for element
concentration) were excluded from the dataset.
**Petrological characteristic field.** The petrological characteristic fields encompass information on
lithology, depositional facies, and metamorphic grade, which provide essential contextual support for
subsequent isotopic geochemistry analyses. (1) Lithology: The Lithology field records the original
descriptions provided by authors, using terms such as "black shale" "mudstone" "limestone" and
"breccia". The LithType field classifies these lithologies into broader categories, primarily as carbonate
and siliciclastic (Tucker and Wright, 2009), with minor entries for phosphorite and iron formations. (2)
Metamorphic Grade: The metamorphic grade field reflects the extent of metamorphism the samples
have undergone, based on original terminology whenever possible. Common terms include specific
metamorphic facies (e.g., amphibolite, greenschist) as well as general descriptors like
"unmetamorphosed" and "low grade". For Cenozoic samples, which are generally assumed to have
undergone minimal metamorphic alteration (Winter, 2014), any entries lacking detailed descriptions are
uniformly designated as "unmetamorphosed". (3) Depositional Setting: This field records the
depositional environment of each sample, with terms like "neritic" "peritidal" "slope" and "abyssal"
preserved from the original literature. For many ocean drilling samples, depositional settings are
inferred from WaterDepth: depths of 500–2000 m are classified as "bathyal" and depths exceeding
2000 m are designated as "abyssal".
**Data collection sources.** Data in the database primarily originate from published literature and
are traceable via DOI. Some data come from public databases such as PANGAEA, SGP, and NOAA.
Each record includes multiple fields for source information, such as first author, publication year,
article title, reference, DOI, and data source. Metadata fields have been standardized and cleaned via
code to ensure consistency and machine readability, removing special characters while retaining
complete citation formats. This structure allows users to trace data provenance, with DOI or Reference
fields facilitating direct searches on Crossref for verification.
**4 Technical validation**
The DSMS-NI database has undergone meticulous curation and quality control (QC) to ensure data
accuracy, consistency, and scientific value. Each record includes comprehensive metadata to support
traceability and verification. While each entry and significant metadata contain a simple remarks field
(excluded from the main database to prevent clutter), it notes the source or reason for inclusion,
facilitating validation and cross-checking by the data management team. We implemented several QC
measures to verify database accuracy.
**Geographic coordinate verification.** Latitude and longitude values were checked to confirm
they fall within the valid ranges of −90 to 90 and −180 to 180, respectively. Sample coordinates were
cross-referenced with country names and public national boundaries to ensure geographic accuracy.
Modern sample coordinates were projected onto a global map with administrative boundaries (Figs. 2-3)
to verify logical placements. If coordinates appeared on land or in other unexpected locations, each
entry was manually reviewed and corrected as needed.

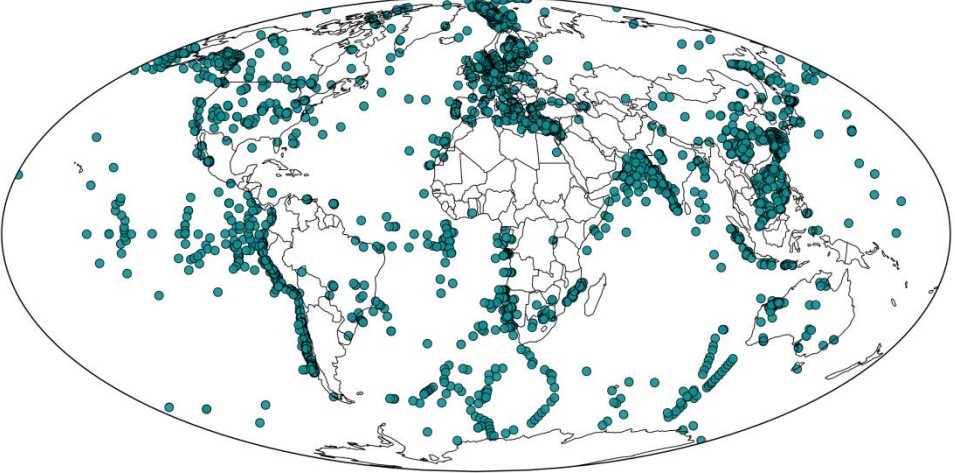


**Figure 2.** Distribution of sample sites on modern global map.

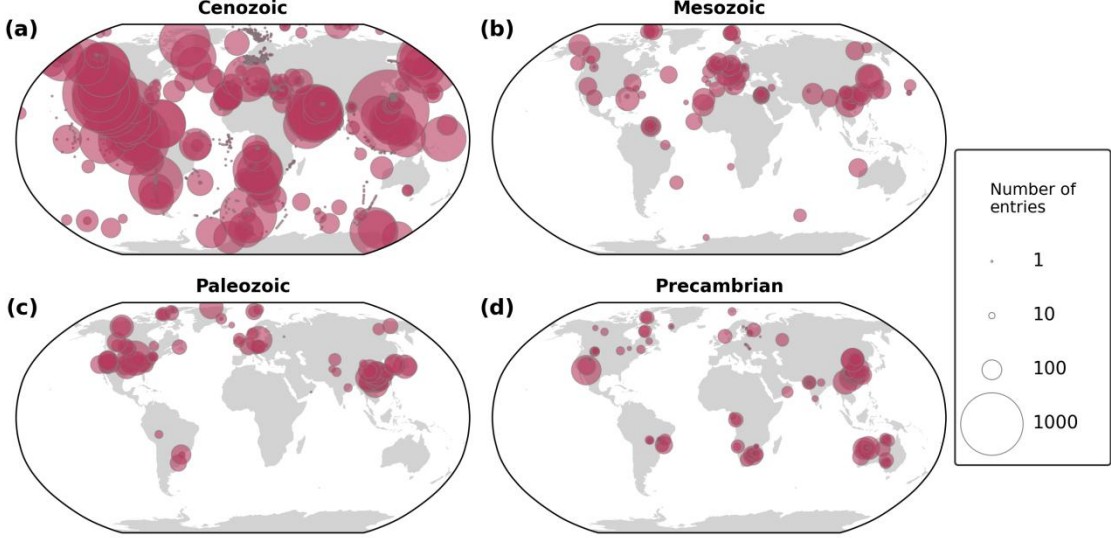

**Figure 3.** Spatial distribution of sampling sites and sample quantities by geological era in a modern geographic reference frame. The base map is adapted from Kocsis and Scotese (2021). The term "entries" refers to individual proxy values and their associated metadata (i.e., rows in the DSMS-NI database).

**Paleocoordinate validation.** Paleolatitudes and paleolongitudes were calculated using the G-Plates model (Scotese and Wright, 2018) and PointTracker v7.0 software, ensuring alignment with each sample's geological age and geographical context. Site locations were plotted on paleogeographic maps (Fig. 4) for further evaluation; any inconsistencies in paleocoordinates were flagged, reviewed, and adjusted accordingly.

**Outlier detection.** Frequency histograms and time-series scatter plots were generated to identify potential outliers in the dataset. All flagged extreme values underwent secondary validation against their original sources to confirm the accuracy. This process led to the correction of erroneous entries introduced during unit conversions and the removal of invalid data points that fell outside instrumental detection limits. Extreme $\delta^{15}N$ values falling outside a conservative range ($< -10$ ‰ or $> +40$‰) were excluded from the final compilation (e.g., Thomazo et al., 2011; Hammarlund et al., 2019). This decision was based not on the validity of the individual measurements, but on the need to prioritize data representativeness for global-scale analysis. The excluded values, even if explained within their original publicational context, are statistical outliers that have not been corroborated and could unduly influence broad interpretations.

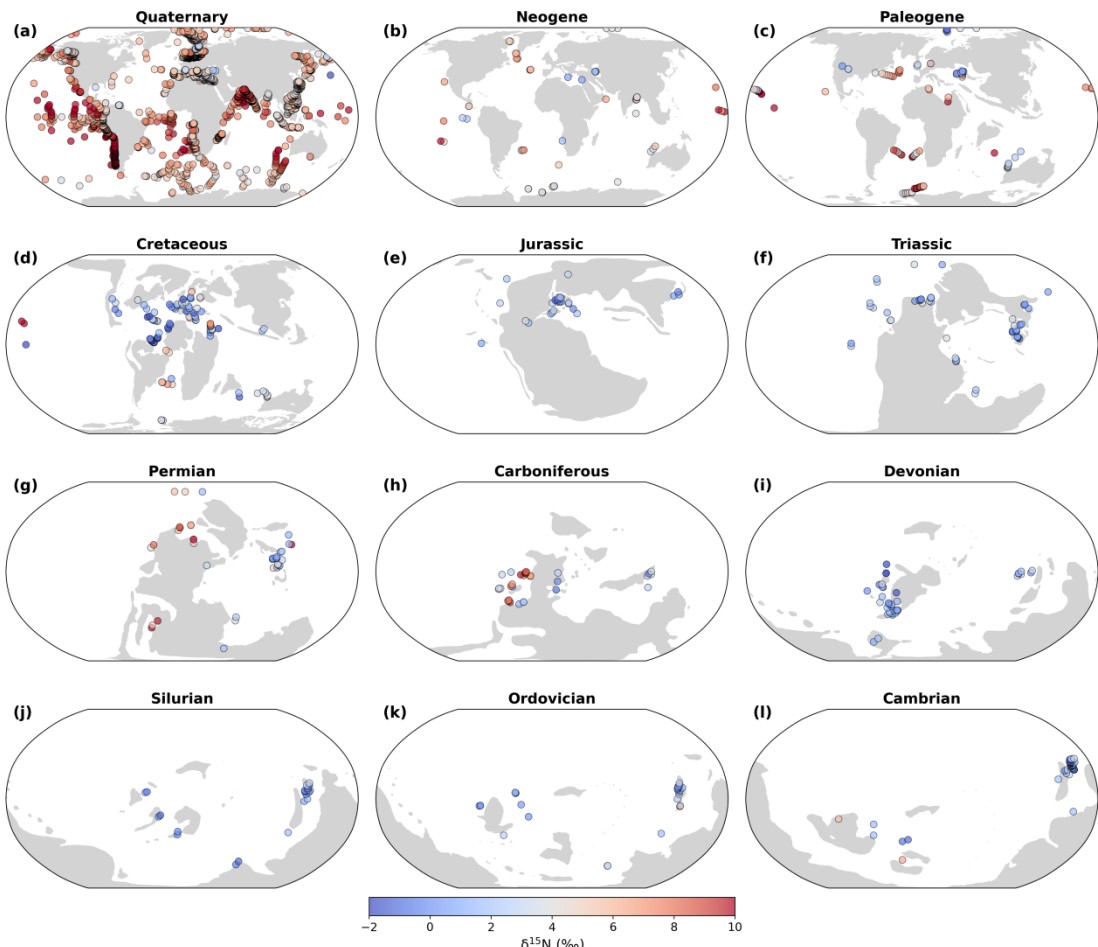

**Figure 4.** Paleogeographic distribution of $\delta^{15}N$ values by geological period. The base map is adapted from Kocsis and Scotese (2021).

**Duplicate check.** We conducted a comprehensive check for duplicate entries, especially for samples with similar GPS coordinates. All suspected duplicates were carefully compared, and necessary corrections were made to eliminate redundancy.

**Age model calibration.** To minimize errors, geological age data were entered using a standardized template to prevent typos, inconsistencies, or incorrect values. Automated analyses and cross-verification ensured that numerical ages corresponded accurately with designated eras and geological stages. A mismatch between a numerical age and its geological stage often indicates an outdated age in the original reference (e.g., Wang et al., 2013). To address this, we recalibrated the outdated estimations by building new age-depth models based on the current geologic stage boundaries from the International Chronostratigraphic Chart (GTS v202309).

**Data collection sources.** Citation information within the reference field was obtained through automated methods from the CrossRef platform, ensuring uniformity in citation formatting (Hendricks et al., 2020). We used scripts to extract comprehensive bibliographic details for each publication,

including author names, title, publication year, journal name, volume, page numbers, and DOI. This automation significantly reduced potential spelling errors and inconsistencies that may arise in manual entry. Extracted citation data were cross-checked against original entries in the database, and any discrepancies or errors were corrected manually by the data management team to maintain source accuracy and completeness.

**5 General database statistics**

The latest version of the DSMS-NI database comprises approximately 320 000 data entries, including 70 854 $\delta^{15}N$ records, spanning all geological periods from the Eoarchean (~3800 Ma) onward. These records originate from a diverse array of unique sampling sites, encompassing ocean drilling cores and outcrop sections. The $\delta^{15}N$ data are predominantly concentrated in the Phanerozoic, comprising 92.1 % of the total database, with further breakdowns showing 71.7 % in the Cenozoic, 8.3 % in the Mesozoic, and 12.1 % in the Paleozoic (Table 3 and Fig. 3). The following sections focus on first-order spatial and temporal trends in $\delta^{15}N$ data density, sampling locations, and values within DSMS-NI. The provided figures illustrate only a subset of the spatial-temporal patterns uniquely revealed by this extensive compilation, demonstrating the database's potential to advance research in paleoclimate, geochemistry, and paleoecology.

**Table 3** The quantities and proportions for $\delta^{15}N$, $\delta^{13}C_{org}$, TN, and TOC of each geological era.

| Proxy system | Cenozoic | Mesozoic | Paleozoic | Precambrian | Total |
|---|---|---|---|---|---|
| $\delta^{15}N$ | 50 795 | 5877 | 8555 | 5625 | 70 852 |
| | 71.7 % | 8.3 % | 12.1 % | 7.9 % | |
| $\delta^{13}C_{org}$ | 10 783 | 4882 | 6873 | 4494 | 270 32 |
| | 39.9 % | 18.1 % | 25.4 % | 16.6 % | |
| TN | 31 530 | 2852 | 6441 | 4977 | 45 800 |
| | 68.8 % | 6.2 % | 14.1 % | 10.9 % | |
| TOC | 22 615 | 5059 | 8555 | 5118 | 41 347 |
| | 54.7 % | 12.2 % | 20.7 % | 12.4 % | |

**5.1 Temporal density and evolution of δ¹⁵N**

Given that the data are concentrated in the Phanerozoic, for which ages are more precisely constrained, we performed a detailed stratigraphic breakdown of age distribution by stage within the Phanerozoic (Fig. 5). The distribution is uneven, with the highest data densities in recent periods, particularly the Holocene (0–12 ka), Late Pleistocene (12–129 ka), and Chibanian (129–770 ka). The high data density in the Quaternary primarily reflects the abundance of high-resolution records from various ocean drilling projects, whose individual cores contributed large and densely sampled datasets. In contrast, older geological periods exhibit data clusters around key events, such as biotic radiations, mass extinctions, and oceanic anoxic events (Bush and Payne, 2021). Notable gaps or low-density intervals occur in the mid-Cambrian to Early Ordovician, Silurian to Early Devonian, mid-Carboniferous to Early Permian, mid-Triassic to Early Jurassic, and Late Jurassic to Early Cretaceous.

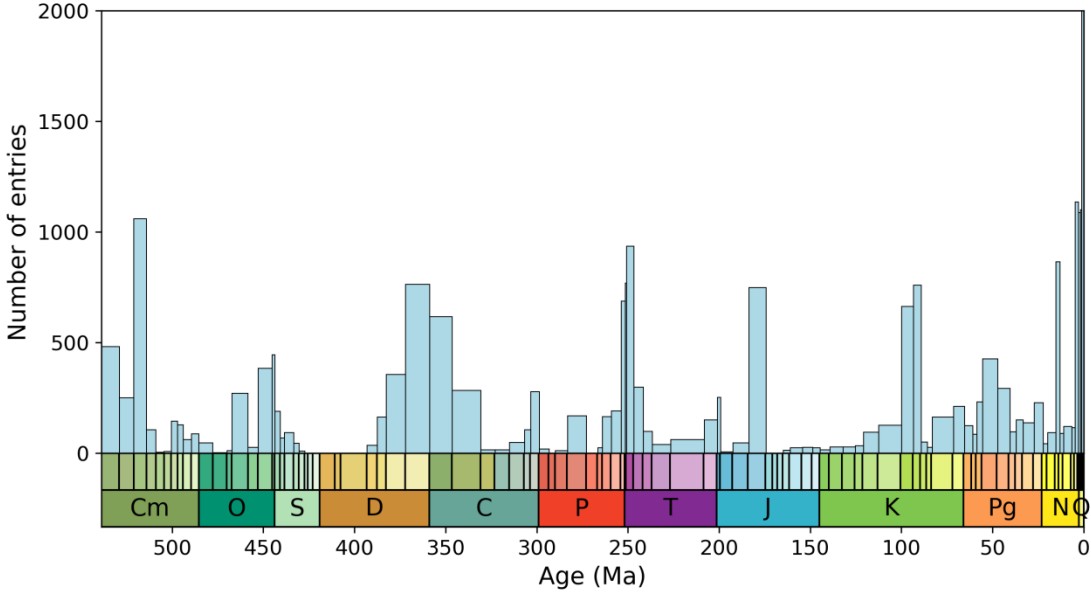

**Figure 5.** Number of data points binned by geologic stage. Data counts for the Holocene (0–12 ka), Late Pleistocene (12–129 ka), and Chibanian (129–770 ka) stages are 10 640, 21 754, and 8378, respectively; these counts are not displayed in the figure due to narrow column width. The Precambrian has only 5646 data points, accounting for 7.9%, and is not shown. Cm: Cambrian; O: Ordovician; S: Silurian; D: Devonian; C: Carboniferous; P: Permian; T: Triassic; J: Jurassic; K: Cretaceous; Pg: Paleogene; N: Neogene; Q: Quaternary.

Overall, δ¹⁵N values exhibit a unimodal distribution centered around +5 ‰, with a mean of 5.1 ± 9.1 ‰ (1σ; Fig. 6a). When examining the modal values of the era-specific kernel density distributions, the Cenozoic exhibits the highest mode, followed by the Precambrian, with significantly lower modal

densities in the Paleozoic and Mesozoic (Fig. 6b). The Precambrian data, which have a dispersed
distribution (Fig. 7), indicate an unstable nitrogen cycle, a state potentially driven by the evolution of
microbial metabolisms and later overprinted by metamorphism (see Ader et al., 2016; Stüeken et al.,
2024 for further discussion). LOWESS smoothing results reveal $\delta^{15}N$ peaks in the Neoarchean,
Paleoproterozoic, and Ediacaran, i.e., periods closely associated with significant oxygenation events
(Kipp et al., 2018; Koehler et al., 2019; Pellerin et al., 2024).

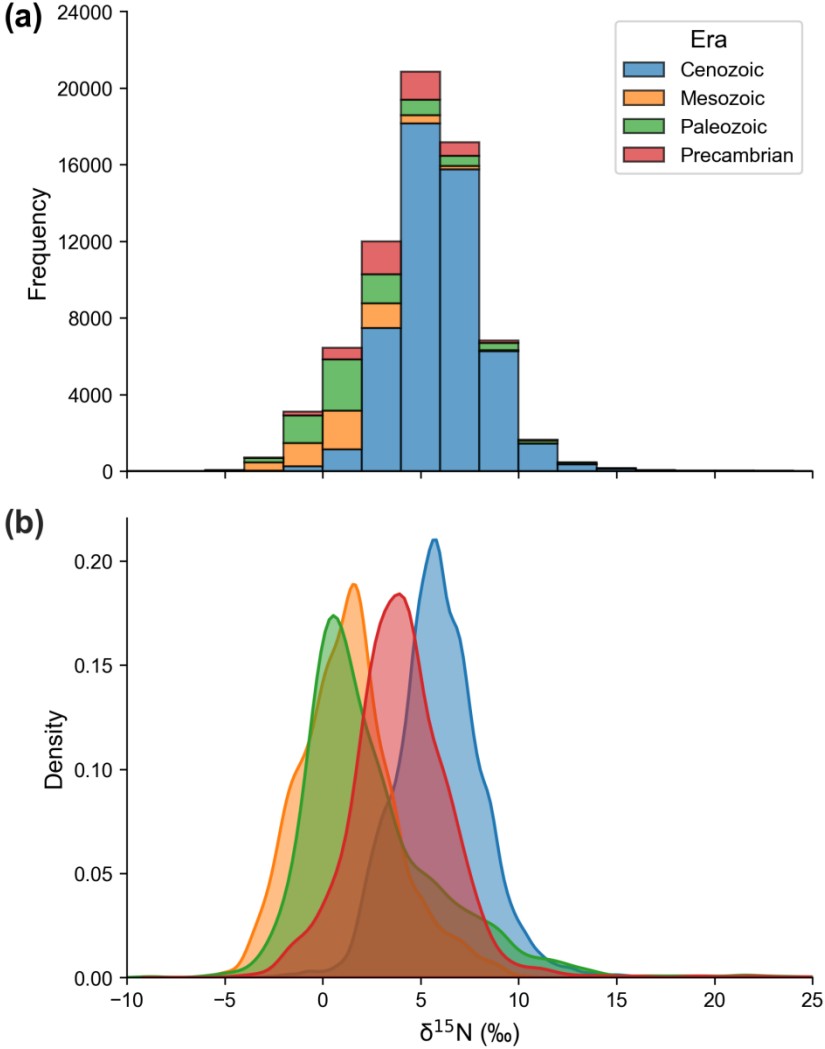


**Figure 6.** (a) Histogram and (b) density distribution of all $\delta^{15}N$ data ($n = 69\ 697$).

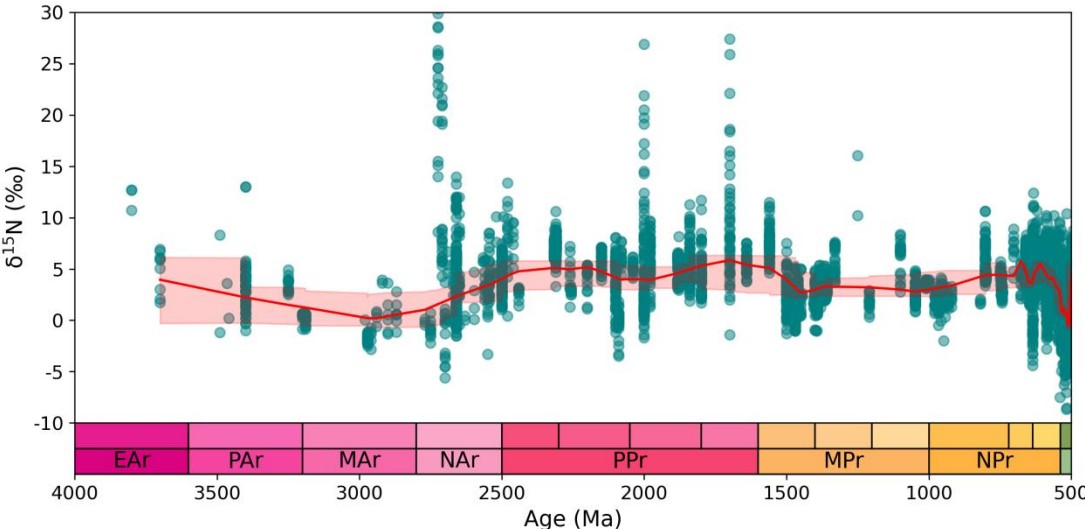


**Figure 7.** δ¹⁵N data and LOWESS curve through Precambrian. A LOWESS factor of 0.01 and a confidence

interval of 2.5–97.5 % were applied. EAr: Eoarchean; PAr: Paleoarchean; MAr: Mesoarchean; NAr: Neoarchean;

PPr: Paleoproterozoic; MPr: Mesoproterozoic; NPr: Neoproterozoic.

An examination of δ¹⁵N record reveals first-order variations on multi-million-year timescales

since the Cryogenian. The LOWESS curve shows extended intervals of relatively elevated δ¹⁵N (>

+5 ‰) during the Cambrian/Ordovician transition, the Carboniferous–Permian, and the late

Cretaceous–Cenozoic (Fig. 7). These broad peaks are separated by periods of lower δ¹⁵N values during

the Ediacaran–Cambrian, Ordovician–Devonian, and Triassic–Cretaceous. The prolonged intervals

(except for the Cambrian/Ordovician transition) of elevated δ¹⁵N broadly coincide with known periods

of sustained cool climates or major glaciations (i.e., the Sturtian–Marinoan glaciations, the Late

Paleozoic Ice Age, and the Cenozoic Icehouse), whereas the low δ¹⁵N intervals generally align with

warmer greenhouse periods (i.e., most of the late Ediacaran–early Carboniferous and the Mesozoic)

(Montañez et al., 2011; Macdonald et al., 2019). This tectonic-scale pattern mirrors observations from

orbital-scale glacial-interglacial cycles (Ren et al., 2017) and transient hyperthermal events like the

Paleocene/Eocene Thermal Maximum (Junium et al., 2018), suggesting that climate exerts a first-order

influence on the marine nitrogen cycle. The underlying mechanisms may involve variations in ice sheet

extent and sea level, which affect the distribution of oxygen-minimum zones (OMZs) and the

proportion of water-column denitrification versus sedimentary denitrification (Algeo et al., 2014; Wang

et al., 2022). However, the correlation is not straightforward; for instance, the increase in δ¹⁵N began in

the Late Cretaceous, coinciding with the onset of global cooling but preceding the major expansion of

Antarctic ice sheets in the Cenozoic (Judd et al., 2024). Therefore, the exact mechanisms coupling

climate and nitrogen cycle evolution remain an unsolved question for future research, ideally

integrating Earth system models with the spatial δ¹⁵N data presented here.

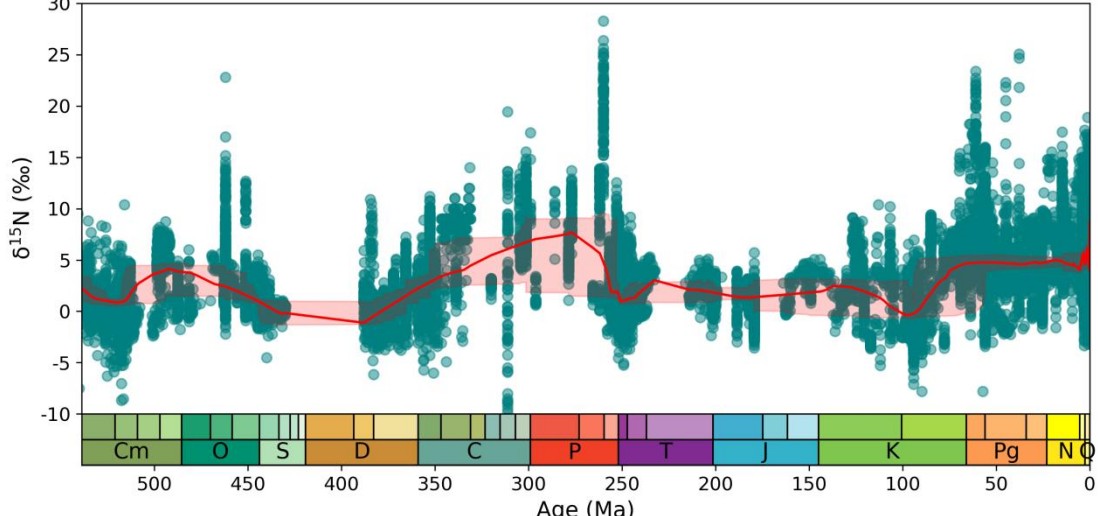

**Figure 8.** δ¹⁵N data and LOWESS curve through the Phanerozoic. A LOWESS factor of 0.03 and a confidence

interval of 2.5–97.5 % were applied.

**5.2 Spatial density and characteristics of δ¹⁵N**

Spatial trends in data density within the DSMS-NI database reveal substantial variability in both

modern (Fig. 2) and paleogeographic distributions (Fig. 4). Ocean drilling sites are primarily

concentrated along continental margins and deep-sea basins, with significant gaps in central oceanic

regions (National Research Council, 2011). For older strata (pre-Cretaceous), sampling sites are

clustered in North America, Europe, China, and South Africa (Fig. 2). In terms of latitude, δ¹⁵N

sampling in older strata is sparse in the modern equatorial region and the mid- to high-latitude areas of

the Southern Hemisphere, aside from some Southern Hemisphere samples collected from Cenozoic

ocean drilling sites (Fig. 3). When modern coordinates are converted to paleolatitudes and mapped onto

paleogeographic reconstructions, the Cenozoic Era provides the most extensive latitudinal coverage,

with the Quaternary period contributing the highest number of sites, followed by the Cretaceous (Figs.

4 and 9). In terms of marine spatial distribution, δ¹⁵N data since the Cretaceous reflects global patterns

to a certain degree (Fig. 4). However, pre-Jurassic data remain spatially concentrated, with Paleozoic

sites limited to just two or three main areas. High-latitude sampling is generally scarce, with Paleozoic

sites predominantly in the Southern Hemisphere and Mesozoic sites mainly in the Northern

Hemisphere (Fig. 4).

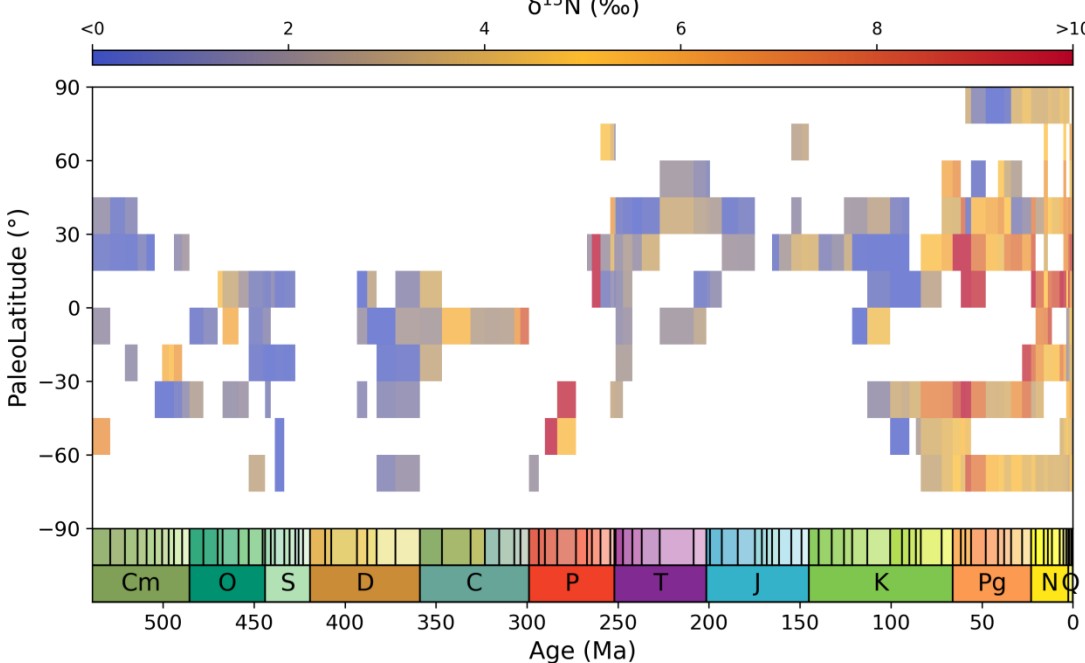

443

**Figure 9.** Spatio-temporal trends in δ¹⁵N values through the Phanerozoic, binned and averaged temporally by stage and spatially by 15° paleolatitudinal bins.

446

To visualize spatial trends, average $\delta^{15}N$ values from each Phanerozoic period were mapped onto paleogeographic reconstructions for the respective period (Fig. 4). Significant spatial differences exist in $\delta^{15}N$ distribution for different geological periods. In modern ocean sediments, elevated $\delta^{15}N$ values (notably > +5 ‰) are concentrated in regions influenced by upwelling, such as the Arabian Sea, southeastern Indian Ocean, eastern equatorial Pacific, southwestern South America, and the western coast of Mexico (Fig. 4a; Tesdal et al., 2013; Du et al., 2005b). In contrast, lower $\delta^{15}N$ values (significantly < +5 ‰) are typically found in restricted basins or broad continental shelves, such as the Black Sea, the Mediterranean Sea, the Baltic Sea, and the South China Sea. The global mean $\delta^{15}N$ (approximately +5 ‰, as observed in open ocean like the Southern Ocean) lies between these extremes. The modern spatial distribution of $\delta^{15}N$ can provide a valuable framework for interpreting past marine conditions, as $\delta^{15}N$ serves as an indicator of nutrient supply, upwelling intensity, and the extent of oceanic oxygen minimum zones (Altabet et al., 1999; Godfrey et al., 2025). However, analyzing spatial patterns in deep-time $\delta^{15}N$ records is inherently limited by the scarcity of data, particularly from open-ocean settings, making it difficult to estimate global mean values and relative spatial gradients. For the Paleogene and Neogene, $\delta^{15}N$ values were generally higher in the open ocean than in continental margin and restricted basins (Fig. 4b-c). In the Paleozoic and Mesozoic, $\delta^{15}N$ values are generally negative, lacking prominent hotspots except in the Carboniferous and Permian. This pattern

may reflect a systematic bias, as available data are predominantly derived from continental shelf
environments (Judd et al., 2020), which tend to exhibit lower $\delta^{15}N$ values compared to the open ocean.
Despite differences in paleogeographic position and absolute $\delta^{15}N$ values, rapid shifts in $\delta^{15}N$ exhibit
consistent directional changes (increase or decrease) during key Phanerozoic transition events, such as
the Permian-Triassic boundary (Knies et al., 2013; Du et al., 2021, 2023) and the Late Cretaceous
(Meyers et al., 2009; Junium et al., 2018; Du et al., 2025b). Given the current uneven distribution of
sampling sites, further $\delta^{15}N$ studies of multiple regions are crucial for enhancing our understanding of
the spatial characteristics of nitrogen cycle evolution in deep time.
**6 Usage notes**
**6.1 Informed user notice**
Each record (row) in the database includes detailed temporal and spatial metadata, along with lithology,
metamorphic grade, and depositional facies information, where available. These metadata are essential
for evaluating the geological context and fidelity of nitrogen isotope data. However, this version of the
database has certain limitations: it may not capture all possible geological age uncertainties or precise
depositional environment details for some records; significant gaps remain in the compilation of data
for certain materials and time intervals (e.g. Quaternary). Consequently, users may need to
independently assess and refine the metadata (e.g., chronological constraints) and supplement missing
data (e.g., coral-bound $\delta^{15}N$ records) as necessary for their specific applications. Despite our extensive
efforts to accurately identify and quality-control each entry, given the vast dataset, some overlooked
errors or data inconsistencies may remain. Users are encouraged to report any issues or omissions to
the authors, as corrections will be incorporated into future database versions. We plan to release a new
version of the dataset annually on Zenodo and update it on the Geobiology Database website. Each
version will incorporate corrections to identified errors and integrate newly published data from the
previous year to the fullest extent possible. This systematic update cycle is designed to ensure the
dataset's accuracy, relevance, and long-term value for the research community.

489       In addition to $\delta^{15}N$ data, the database provides geochemical information such as TOC, total TN,

$\delta^{13}C_{org}$, and major and trace element concentrations. These supplementary data are valuable for
assessing factors that may influence nitrogen isotopes, such as organic matter preservation and redox
conditions. Even when not directly paired with $\delta^{15}N$ values, we retain all relevant data to enable users
to conduct correlation analyses via interpolation or other methods. Researchers are welcome to
contribute additional geochemical data from the same sites or samples as they become available,
allowing for updates and refinements in subsequent database releases.
**6.2 Applying the database to deep-time studies**
When applying the database to deep-time studies, certain filtering criteria can be used. For instance,
samples may be selected based on lithology, metamorphic grade, and other metadata to ensure that the
data aligns with specific geological research contexts. Temporal, paleolatitude, and paleodepth
information are critical for paleogeographic reconstructions and spatiotemporal distribution analyses,
particularly when investigating paleoclimate change and global biogeochemical cycles. Further
analysis of variations in latitude, basin characteristics, and water depth has the potential to yield
significant insights. Given the rapid variability of nitrogen isotopes and their pronounced regional
characteristics, filling temporal and spatial gaps and enhancing resolution are of great
value—particularly for pivotal periods like the Ordovician-Silurian mass extinction, the Early
Devonian terrestrial plant radiation, and the Late Jurassic-Early Cretaceous supercontinent breakup.
The database is also especially suited for comparative studies of key geological periods, such as the
Permian-Triassic boundary extinction and the Cretaceous OAE2. Given the inherent limitations of our
simplified age-depth models, we recommend that users seeking higher chronological precision for
time-series analysis incorporate additional stratigraphic constraints (e.g., paleomagnetic or
cyclostratigraphic data) to develop finer-scale age models, where necessary. To support these
applications, we have also provided a software tool on Zenodo, allowing users to generate heatmaps of
$\delta^{15}N$ data distributions for specific time intervals. These heatmaps visualize the average spatial
distribution of $\delta^{15}N$ for any selected geological interval, offering preliminary validation for user
hypotheses and aiding in uncovering the evolution of the global nitrogen cycle.
**7 Data availability**
The DSMS-NI version 0.3 can be accessed via Zenodo at https://doi.org/10.5281/zenodo.15117375 (Du
et al., 2025a) and via the GeoBiology website at https://geobiologydata.cug.edu.cn/ (last access: April

519     30 2025).

**8 Code availability**
The code used to validate the dataset, make the figures in this manuscript, and the heatmap tool is

available on Zenodo (https://doi.org/10.5281/zenodo.15758073). The paleocoordinates were estimated using the PointTracker v7 tool published by the PALEOMAP Project, which can be found at http://www.paleogis.com (last access: April 1 2025; https://doi.org/10.13140/RG.2.1.2011.4162, Scotese, 2008).

**Author contributions.** YD, HYS and HJS designed the study and secured funding. TJA, EES, SEG, JDC, YD, HZ, XKL, JP, YW, JK, XS, HS, DC and LT conducted data acquisition, curation and validation. YD, LW, JZ, QL, XCL and HY developed computational methodologies and provided technical support. YD prepared the paper with contribution from all co-authors.

**Competing interests.** The authors declare that they have no conflicts of interest.

**Acknowledgements.** This work has benefited from previous $\delta^{15}N$ datasets compiled by Jan-Erik Tesdal, Christophe Thomazo, Thomas J. Algeo, Eva E. Stüeken, Magali Ader, Xinqiang Wang, and Michael A. Kipp.

**Financial support.** This work has been funded by the State Key R&D Project of China (2023YFF0804000), the National Natural Science Foundation of China (42172032; 42325202; 42402316), the China Postdoctoral Science Foundation (2025T180107), the Postdoctoral Fellowship Program of CPSF (GZB20230679), the Postdoctor Project of Hubei Province (2024HBBHCXB087), the Natural Science Foundation of Hubei Province (2023AFA006).

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
