# Peer review of "The global Deep-time Sediment Nitrogen Isotopes in 2 Marine Systems (DSMS-NI) database"

_Earth System Science Data, 2025_

## Author Response (AR1)

**Reply to Reviewer #1' comments**

General comments

Du et al. have compiled an extensive database for marine nitrogen isotopes, covering data dated back to Archean. The database organizes data from over 400 publications and also makes use of existing geochemical databases. Building on existing published data compilations, this deep time marine nitrogen isotopes dataset covers a longer time span, centralizes a range of geochemical proxies and works to provide age estimates that are convenient to use. The authors assembled a total of 71040 δ15N data and 285715 geochemical data that allow quality evaluation and paleoenvironmental interpretations. This database is also accessible to all and the authors plan to continue improving and enlarging the repository upon future research.

This paper goes through data compilation, data summary and validation for data quality. The structure is organized and easy to follow. The authors put into effects to calibrate age estimates that can be used readily and also are in reference to an updated geochronological framework. However, they did not provide sufficient discussion on age uncertainties that their age adjustment might incur. Aside from accessibility of the dataset, the data quality and uncertainty are also essential. Similarly, the discussion on data quality check and validation can benefit from some additional details and explanation (see specific comments).

The other concern I have is the authors' interpretation on database statistics. Although the key objective of this paper is not on paleoenvironmental reconstruction, the authors showcased some new insights the database can bring. I would suggest them to present these findings with interpretation that phrased as proposed or possible mechanism/idea, avoiding making strong arguments for rigor, for example, the variation of δ15N throughout time, and the latitudinal trend of δ15N. In the modern ocean, the spatial pattern of δ15N is not dominantly affected by latitudes. The apparent higher δ15N values at lower latitudes in Cenozoic might be driven by biased focus on regional oceanographic feature, such as the oxygen minimum zones in the tropical Pacific.

I think the presented database would be a beneficial addition for the community studying climate, marine environment and ecology in Earth history. After revisions and clarifying data uncertainties and quality validation, this database can motivate further investigations for understanding the marine environment and biogeochemical cycling throughout Earth history.

**Response:** Thank you very much for your thoughtful and constructive comments regarding our manuscript. We agree with the general comments regarding the importance of clearly addressing age uncertainties, elaborating on data quality validation, and carefully phrasing the interpretations of spatial and temporal δ15N trends. In response, we have enhanced the manuscript by adding additional descriptions regarding age uncertainty and data quality checks. We have also revised the discussion to avoid overinterpretation and have removed the discussion on the

latitudinal gradient changes. Please see our detailed point-by-point responses to the specific comments below.

Specific comments

Line 85: suggest change to "survey of δ15N records on bulk sediments and biominerals deposited within".

**Response:** We have revised the phrase as suggested. See line 82.

Line 96: In Table 1, it would be more clear if having "broad age", "crude age" and "fine age" defined as table annotations or include the definitions in the main text.

**Response:** To improve clarity, we have added the definitions for age resolution in the caption of Table 1. "Broad age" encompasses all data that lack high-resolution chronostratigraphic control. It applies to a single age value in a set of samples, whether it be for an entire geological period or for a specific sedimentary section without an internal age model. "Fine age" is used for data with the high chronological resolution, derived from established age-depth models, which provide sequentially ordered ages.

Line 148: What types of settings are considered highly heterogeneous? It would be good to have a couple of examples. Citation or explanation is needed here.

**Response:** Thank you for your comment. We acknowledge that our original statement was unclear. In our data filtering process, this term was intended to identify samples originating from highly localized environments or those exhibiting extremely high metamorphic grades. For example, some excluded samples are highly metamorphosed minerals (e.g., mica; Jia and Kerrich, 2000; Busigny et al., 2003) or influenced by hydrothermal activity (Martin and Stüeken, 2024). These samples more likely reflect metamorphic processes or hydrothermal alteration rather than marine paleo-environmental conditions. This judgement is based on discussions from the original publications, rather than a strict metamorphic grade threshold. We have revised the text and included explanatory examples for clarity. The revised sentence now reads:

"Data from geological settings representing highly localized environments or with high metamorphic grades, such as highly metamorphosed minerals (e.g., mica; Jia and Kerrich, 2000; Busigny et al., 2003) or samples affected by hydrothermal activity (Martin and Stüeken, 2024), were also excluded. The $\delta^{15}N$ of these samples primarily records alteration processes rather than seawater signatures. This filtering criterion was applied based on descriptions in the original literature rather than a fixed metamorphic grade threshold." (see lines 149-155).

Busigny, V., Cartigny, P., Philippot, P., Ader, M., and Javoy, M.: Massive recycling of nitrogen and other fluid-mobile elements (K, Rb, Cs, H) in a cold slab environment: evidence from HP to

UHP oceanic metasediments of the Schistes Lustrés nappe (western Alps, Europe), Earth and Planetary Science Letters, 215, 27–42, https://doi.org/10.1016/S0012-821X(03)00453-9, 2003.

Jia, Y. and Kerrich, R.: Giant quartz vein systems in accretionary orogenic belts: the evidence for a metamorphic fluid origin from $\delta 15N$ and $\delta 13C$ studies, Earth and Planetary Science Letters, 184, 211–224, https://doi.org/10.1016/S0012-821X(00)00320-4, 2000.

Martin, A. N. and Stüeken, E. E.: Mechanisms of nitrogen isotope fractionation at an ancient black smoker in the 2.7 Ga Abitibi greenstone belt, Canada, Geology, 52, https://doi.org/10.1130/G51689.1, 2024.

Line 159: The citation "Farmer et al., 2021" is not listed in the reference. Also, more suitable citations for the denitrifier-based method are Ren et al., 2012 in Limnology and Oceanography, and Smart et al., 2018 in GCA.

 **Response:** The citation to Farmer et al. (2021) has been removed, and Ren et al. (2012) and Smart et al. (2018) are cited now as suggested.

Line 165: It is unclear what "repeated measurements of the same sample" are. Are these replications of the same sample, or measurements for the same type of samples?

 **Response:** The term "repeated measurements of the same sample" specifically refers to replicate analyses performed on the same sample material (e.g., aliquots of the same powder) to evaluate analytical precision and reproducibility. These are not measurements of different samples of the same type. To clarify, we have revised the sentence as follows:

"For data from the same site but at different depths or lithologies, or for measurements of different components at the same layer (e.g., bulk sediment and decarbonated sediment), or replicate analyses of the same homogenized sample, each entry was recorded separately to accurately capture variability."

Line 169: What are the criteria as reasonable explanations for extreme $\delta 15N$? Are the explanations based on the discussion in the source publications? Please include citation or example here.

**Response:** Our primary criterion for excluding extreme $\delta^{15}N$ values was not the lack of a plausible explanation in the source publication, but rather their lack of representativeness for broader geological patterns.

Upon review, we found that while some source publications did offer reasonable, site-specific explanations for extreme values (e.g., volatilization of $NH_3$, see Stüeken et al., 2015), these values were extreme outliers that have rarely been replicated in other studies from other sites. For a global compilation aimed at identifying broad-scale trends, including such rare, non-reproducible data points could disproportionately skew the statistical analysis and obscure general patterns.

Therefore, we applied a conservative filter, excluding data points with $\delta^{15}N < -10‰$ or $> +40‰$ (e.g., Thomazo et al., 2011; Hammarlund et al., 2019). This threshold is based on the observation that the vast majority of marine sedimentary $\delta^{15}N$ values in

our database fall within the range, which robustly represents the known bounds of biological nitrogen cycling (Sigman and Fripiat, 2019). The excluded extreme values, despite potential local significance, were deemed non-representative at the global scale. We have revised the text and incorporated it into Section 4, 'Technical validation' for better clarity (Lines 323-328):

"Extreme $\delta^{15}$N values falling outside a conservative range (< -10‰ or > +40‰) were excluded from the final compilation (e.g., Thomazo et al., 2011; Hammarlund et al., 2019). This decision was based not on the validity of the individual measurements, but on the need to prioritize data representativeness for global-scale analysis. The excluded values, even if explained within their original publicational context, are statistical outliers that have not been corroborated across different sites and could unduly influence broad interpretations."

Hammarlund, E. U., Smith, M. P., Rasmussen, J. A., Nielsen, A. T., Canfield, D. E., and Harper, D. A. T.: The Sirius Passet Lagerstätte of North Greenland-A geochemical window on early Cambrian low-oxygen environments and ecosystems, Geobiology, 17, 12–26, https://doi.org/10.1111/gbi.12315, 2019.

Sigman, D. M. and Fripiat, F.: Nitrogen Isotopes in the Ocean, in: Encyclopedia of Ocean Sciences, Elsevier, 263–278, https://doi.org/10.1016/B978-0-12-409548-9.11605-7, 2019.

Stüeken, E. E., Buick, R., and Schauer, A. J.: Nitrogen isotope evidence for alkaline lakes on late Archean continents, Earth Planet. Sci. Lett., 411, 1–10, https://doi.org/10.1016/j.epsl.2014.11.037, 2015.

Thomazo, C., Ader, M., and Philippot, P.: Extreme 15N-enrichments in 2.72-Gyr-old sediments: evidence for a turning point in the nitrogen cycle, Geobiology, 9, 107–120, https://doi.org/10.1111/j.1472-4669.2011.00271.x, 2011.

Line 193: In Table 2, under field name "Isotopes", change "the isotope composition" to "the isotopic composition".

 **Response:** Revised.

Line 193: In Table 2, under field name "Reference", change the typo "formated" to "formatted".

 **Response:** Revised.

Lines 234-237: The explanation for age adjustment here seems to be repetitive as explained in the "Age model calibration" (Lines 319-322).

  **Response:** We thank the reviewer for pointing out the repetition. To avoid redundancy, the explanatory sentences previously in Lines 319-322 regarding the construction of age-depth models and assignment of median ages have been removed from the "Age model calibration" section.

Line 308: An expected δ15N range needs to be defined here, as what δ15N values would be considered "unusually high or low"? If so, do the outlier δ15N get removed from the database? What validation process or criteria that determine whether the data is included or not?

**Response:** We have revised the text to explicitly define the threshold and clarify our validation process for data fields. The text now reads as follows in the manuscript (Lines 320-328):

"All flagged extreme values underwent secondary validation against their original sources to confirm the accuracy. This process led to the correction of erroneous entries introduced during unit conversions and the removal of invalid data points that fell outside instrumental detection limits. Extreme $\delta^{15}N$ values falling outside a conservative range (< -10‰ or > +40‰) were excluded from the final compilation (e.g., Thomazo et al., 2011; Hammarlund et al., 2019). This decision was based not on the validity of the individual measurements, but on the need to prioritize data representativeness for global-scale analysis. The excluded values, even if explained within their original publicational context, are statistical outliers that have not been corroborated and could unduly influence broad interpretations."

To directly address these questions:

(1) Definition of "unusually high or low $\delta^{15}N$": We have now explicitly defined the outlier range as $\delta^{15}N$ < -10‰ or > +40‰. This conservative threshold is based on the observed distribution within our own extensive compilation of geological records.

(2) Removal from database: Yes, data points identified as outliers based on this criterion were excluded from the final dataset. It should be noted that only a very small number of $\delta^{15}N$ was excluded.

(3) Validation criteria: For $\delta^{15}N$ data, our primary inclusion criterion was data accuracy and consistency with the global range of $\delta^{15}N$ values in geological records, ensuring the integrity of our subsequent interpretations. For element data, we performed a systematic correction of erroneous entries, including unit errors and values exceeding reported detection limits; these data were either corrected or removed.

Lines 319-322: Does the adjustment for outdated age estimations only refer to the boundaries between geologic stages of the International Chronostratigraphic Chart? Also, it is unclear that how the age-depth model is developed, assuming a constant sedimentation rate? Were any other geochronological tie points considered for the age-depth model, such as the paleomagnetic reversals? Adjusting the age with "the median age of the corresponding geologic interval" might introduce age uncertainties. I suggest the authors to add some discussion or comments on the age uncertainties, which would provide more rigorous quality evaluation for users in the future.

**Response:** We thank the reviewer for these insightful comments regarding the age models, which are crucial for interpreting our compiled dataset. We have revised the

manuscript to provide greater clarity and to address the concerns about age uncertainties as suggested.

To directly address these questions:

(1) Clarification on the adjustment for outdated age estimations

Yes, the adjustment primarily refers to updating the numerical ages of geologic stage boundaries to align with the recent International Chronostratigraphic Chart. When original publications used outdated chronostratigraphy, we replaced the ages by applying a new age-depth model based on the current geologic stage boundaries from the International Chronostratigraphic Chart.

(2) Methodology of the age-depth model and use of tie points

For constructing the age-depth model, we assumed a constant sedimentation rate between known age tie-points or stage boundaries and assigned sample ages via linear interpolation. We acknowledge that this approach may introduce considerable error in absolute age estimation; however, it primarily serves to preserve the relative sequence among samples and to avoid arbitrary clustering of data at single time horizons.

We fully agree that incorporating other stratigraphic age constraints, such as paleomagnetic reversals and cyclostratigraphy, would significantly enhance age precision. However, such high-resolution data are not consistently available for the vast majority of the deep-time records in our global compilation, and require extensive effort for each section. Manually re-interpreting each section was beyond the scope of this compilation-based study. Therefore, our approach represents a pragmatic and consistent methodology applied across a heterogeneous dataset. We strongly recommend future users recalibrate ages by applying other geochronological tie points if higher resolution is needed.

(3) Discussion of age uncertainties

We have added a dedicated discussion on age uncertainties and recommend that future users conduct additional verification where accurate temporal constraints are essential.

The following revisions were implemented (Lines 339-342, 235-250, 508-511):

"A mismatch between a numerical age and its geological stage often indicates an outdated age in the original reference (e.g., Wang et al., 2013). To address this, we recalibrated the outdated estimations by building new age-depth models based on the current geologic stage boundaries from the International Chronostratigraphic Chart (GTS v202309)."

"(1) For records with at least two samples or stratigraphic horizons of known age (e.g., radiometrically dated layers or well-defined stage boundaries), we constructed an age-depth model. This model linearly interpolates ages between these tie-points along the RelativeDepth axis, assuming a constant sedimentation rate within each interval. The principal simplification of the age-depth model is the assumption of a constant sedimentation rate between stratigraphic age tie-points. While this assumption is

effective for establishing the relative temporal sequence of samples, which is critical for capturing first-order stratigraphic trends, it necessarily introduces uncertainties in absolute age determination due to potential variability in sedimentation rates or local stratigraphic features. (2) For records lacking sufficient data for an age-depth model, a single age was assigned to all samples. When only one age constraint (e.g., a radiometric date from a nearby stratum) is available, that specific age is applied. In the absence of any direct age control, the median age of the corresponding geologic stage is used as a default. It should be noted that assigning a uniform age to a suite of samples, particularly using the median stage age, carries significant uncertainty, theoretically on the order of the duration of the entire geologic interval (which can approach 100 Ma for long stages in the Precambrian). Profiles constrained by a single radiometric date, which is the predominant method for dating sequences older than 600 Ma, are generally more reliable than those relying solely on a median stage age.”

“Given the inherent limitations of our simplified age-depth models, we recommend that users seeking higher chronological precision for time-series analysis incorporate additional stratigraphic constraints (e.g., paleomagnetic or cyclostratigraphic data) to develop finer-scale age models where necessary. ”

Line 367-368: The sentence “the Cenozoic has the highest overall peak” is unclear. Does it refer to the highest peak density? Or do you mean the Cenozoic has the highest mode $\delta^{15}$N value? The sentence “lower peaks in the Paleozoic and Mesozoic” has the same issue.

 **Response:** The “peak” indeed refers to the highest density of data points (i.e., the mode) in the kernel density plot, not the absolute $\delta^{15}$N value. We have revised the sentence as follows to clarify this:

“When examining the modal values of the era-specific kernel density distributions, the Cenozoic exhibits the highest mode, followed by the Precambrian, with significantly lower modal densities in the Paleozoic and Mesozoic.”

Lines 369-371: Does the peak $\delta^{15}$N occur in the Late Cretaceous or at the K/Pg boundary? Also, I did not observe any notable $\delta^{15}$N peaks in mid-Triassic, Jurassic, and Early Cretaceous from Fig. 7. The interpretation that $\delta^{15}$N values “align with greenhouse-icehouse climate cycles” might be too strong, since no correlation is shown for the analysis in the paper. Climate cycles imply that repeated up-and-down feature can be observed, whereas the $\delta^{15}$N record through Phanerozoic does not show clear cyclic pattern.

 **Response:** We thank the reviewer for these critical comments. We apologize for the earlier overstatement and imprecision in the text. The intended point was that the broad, long-term shifts in the $\delta^{15}$N record appear to correspond with the major climatic states of the Phanerozoic, rather than implying short-term, cyclic behavior. We have revised and expanded this section to clarify the temporal patterns observed in our $\delta^{15}$N dataset and their potential link to climate (Lines 403-422):

"An examination of δ15N record reveals first-order variations on multi-million-year timescales since the Cryogenian. The LOWESS curve shows extended intervals of relatively elevated δ15N (> +5 ‰) during the Cambrian/Ordovician transition, the Carboniferous–Permian, and the Late Cretaceous–Cenozoic (Fig. 7). These broad peaks are separated by periods of lower δ15N values during the Ediacaran–Cambrian, Ordovician–Devonian, and Triassic–Cretaceous. The prolonged intervals (except for the Cambrian/Ordovician transition) of elevated δ15N broadly coincide with known periods of sustained cool climates or major glaciations (i.e., the Sturtian–Marinoan glaciations, the Late Paleozoic Ice Age, and the Cenozoic Icehouse), whereas the low δ15N intervals generally align with warmer greenhouse periods (i.e., most of the late Ediacaran–early Carboniferous and the Mesozoic) (Montañez et al., 2011; Macdonald et al., 2019). This tectonic-scale pattern mirrors observations from orbital-scale glacial-interglacial cycles (Ren et al., 2017) and transient hyperthermal events events like the Paleocene/Eocene Thermal Maximum (Junium et al., 2018), suggesting that climate exerts a first-order influence on the marine nitrogen cycle. The underlying mechanisms may involve variations in ice sheet extent and sea level, which affect the distribution of oxygen-minimum zones (OMZs) and the proportion of water-column denitrification versus sedimentary denitrification (Algeo et al., 2014). However, the correlation is not straightforward; for instance, the increase in δ15N began in the Late Cretaceous, coinciding with the onset of global cooling but preceding the major expansion of Antarctic ice sheets in the Cenozoic (Judd et al., 2024). Therefore, the exact mechanisms coupling climate and nitrogen cycle evolution remain an unsolved question for future research, ideally integrating Earth system models with the spatial δ15N data presented here."

Lines 411-413: The Fig. 9 does not include data for the Ediacaran. The citation of "Fig. 9" should move to the first half sentence.

 **Response:** Revised.

Database file (DSMS-NI_v0.2): Under the column for "Material", a few options refer to the same thing – foraminifer, foraminifera, foraminifers, and planktic foraminifera. All of these data are foraminifera-bound organic matter nitrogen isotopes. Different taxa of planktic/planktonic foraminifera might be used for analysis in these source publications. It could be very useful to clarify the taxonomic information.

 **Response:** We have standardized them to "foraminifera" to ensure consistency and added taxonomic information following the term "foraminifera".

**Reply to Reviewer #2' comments**

General Comments

Du et al. have established the Database of Deep-time Sediment Nitrogen Isotopes in Marine Systems (DSMS-NI) in this work. By integrating previously published datasets and supplementing them with newly collected data, this dataset represents a comprehensive compilation in the field of marine sediment nitrogen isotopes. It includes 71,040 δ15N data points spanning from the Archean to the modern, and covers a wide range of sample types, from fossil materials to bulk sediments, kerogen, and others. The authors have also made substantial efforts to enrich each nitrogen isotope data point with additional metadata, such as modern latitude and longitude, paleolatitude and paleolongitude, age information, lithology, depositional water depth, and, where available, other geochemical information from the same stratigraphic horizon. This endeavor greatly expands the potential for future research based on this database.

**Response:** We sincerely appreciate your positive feedback and constructive comments.

At the same time, several issues should be addressed to make the dataset more robust and user-friendly:

1. Although the manuscript carefully explains the metadata fields, the database itself is not accompanied by a Data Descriptor File (ReadMe file). Providing such a file would substantially improve data usability and accessibility for the community.

**Response:** A ReadMe file is now attached to the database.

2. In the current dataset, materials are categorized into sediments, foraminifera, diatoms, etc. However, foraminifera- and diatom-bound nitrogen isotope measurements are reported as bulk δ15N values, while porphyrin and kerogen are listed separately. This classification seems inconsistent and somewhat confusing. A clearer approach might be to keep "sediments" and sedimentary rock types as the main material categories, and then include foraminifera, diatoms, porphyrin, and kerogen as specific entries in the data fields.

**Response:** We have restructured the database into a two-tiered classification system and revised the corresponding descriptions in the manuscript to implement a clearer and more logical framework. The two-tiered classification includes:

(1) The term $\delta^{15}N_{bulk}$ is now exclusively used for measurements performed on bulk sediment or decarbonated sediment. These are now clearly marked as such in the "Material" column.

(2) A new category, $\delta^{15}N_{sp}$, has been established to classify measurements targeting specific phases within a sample, such as foraminifera, diatom, porphyrins, kerogen, and clay-bound nitrogen. These are also distinctly marked in the "Material" column.

3. While the dataset is comprehensive, it does not appear to include coral-bound δ15N records, which have become an increasingly important proxy for reconstructing Holocene nitrogen cycling. Incorporating, or at least acknowledging, this data type would further strengthen the completeness of the compilation.

**Response:** We thank the reviewer for this valuable suggestion regarding coral-bound $\delta^{15}N$ records. The current version of our compilation indeed has a primary focus on the deep-time record, where data are severely underrepresented. We fully agree that incorporating coral-bound $\delta^{15}N$ data is crucial, particularly for assessing Holocene nitrogen cycling and isotopic fractionation in biogenic carbonates. We have acknowledged this limitation in Section 6.1 Informed user notice, and plan to include them in the next version of the database.

4. The dataset would benefit from including a field specifying the analytical method used for δ15N measurements. Different techniques (e.g., EA flash combustion, denitrifier method, chemical oxidation, offline combustion) may introduce distinct biases, and distinguishing these would be important for future analyses.

**Response:** We have added a new column titled "Technique$\delta^{15}N$" to the dataset. This column now compiles the analytical method for each record, such as EA combustion, offline combustion, and denitrifier method.

5. Some citation and metadata errors are present. For example, data attributed to Alt-Epping U. et al. (2009) shows negative ages, and some entries cite conference abstracts (e.g., Wang X. et al. 2021, Goldschmidt 2021 abstracts) rather than the final peer-reviewed sources. In addition to automated quality control, it would be helpful to perform manual spot-checking of a representative subset of entries against the original literature. If the authors could demonstrate that the error rate is low, this would increase confidence in the dataset.

**Response:** We are grateful to the reviewer for their careful review and for highlighting these metadata errors. We have addressed them as follows:

(1) Negative Ages (e.g., Alt-Epping et al., 2009): we have corrected two data points. The error occurred because ages in the Common Era should be set to zero, but two entries were mistakenly overlooked during initial compilation.

(2) Citation Errors: we have performed a thorough manual check of all reference sources. We found that the initial errors were primarily due to inaccuracies in automated matching via Crossref. A thorough manual verification of all references has been completed to ensure data integrity; for example, a preprint (Godfrey et al., 2024) has recently been updated to a peer-reviewed publication (Godfrey et al., 2025).

Godfrey, L., Omta, A. W., Tziperman, E., Li, X., Hu, Y., and Falkowski, P.: The marine nitrogen cycle over the past 165 million years, https://doi.org/10.21203/rs.3.rs-3417349/v1, 26 January 2024.

Godfrey, L. V., Omta, A. W., Tziperman, E., Li, X., Hu, Y., and Falkowski, P. G.: Stability of the marine nitrogen cycle over the past 165 million years, Nat. Commun., 16, 8982, https://doi.org/10.1038/s41467-025-63604-x, 2025.

6. It would be helpful if the authors could clarify their plans for future updates and maintenance of DSMS-NI, and whether mechanisms will be provided for incorporating newly published data or community-submitted contributions. This would further increase the long-term value and sustainability of the dataset.

 **Response:** We have established the following plans for updates and maintenance:

(1) The corresponding and first author will be responsible for curating and releasing an updated version of the complete dataset on Zenodo at the end of each year. Each new version will incorporate corrections to any identified errors, include necessary metadata revisions, and integrate newly published data from the preceding year. Zenodo provides Digital Object Identifiers (DOIs) for each version, which will ensure citability and traceability of specific dataset releases.

(2) The dataset is hosted and will be updated on the Geobiology Database website. This platform, which is currently under active development to add new features and new database (e.g., sulfur isotopes), will support more frequent and incremental updates.

We have now explicitly described these maintenance and update plans in the "Usage Notes" section (Section 6):

"We plan to release a new version of the dataset annually on Zenodo and update on the Geobiology database website. Each version will incorporate corrections to identified errors and integrate newly published data from the previous year to the fullest extent possible. This systematic update cycle is designed to ensure the dataset's accuracy, relevance, and long-term value for the research community."

In addition, I agree with Anonymous Referee #1 that while the descriptive statistics on the dataset are useful, some of the interpretations presented in the manuscript may be too strong at this moment.

Overall, this manuscript makes an important contribution to the community, and I support its publication after the authors address the above points.

 **Response:** We have carefully revised the interpretation of $\delta^{15}N$ data in Section 5 "General database statistics", as suggested by Referee #1, to ensure that our conclusions are well-supported and appropriately cautious. See line 403-422, 449-471.

[revised manuscript text omitted]

Specific Comments

• Figure 4: Some plotted points appear to fall on land, especially for the Cretaceous interval; please check the basemap.

 **Response:** The points between South America and Africa that appear on land are accurate and correspond to oceanic samples from the Late Cretaceous, a period of major continental drift. To improve clarity, we have revised the figure by adopting a paleogeographic basemap that accurately represents the separated positions of South America and Africa in the Late Cretaceous.

• Lines 374–376: This sentence lacks sufficient supporting evidence or a figure. Please clarify.

 **Response:** This sentence aimed to synthesize a pattern observed across multiple published studies, rather than figures in this manuscript. We have revised the sentence as follows:

"Despite differences in paleogeographic position and absolute $\delta15N$ values, rapid shifts in $\delta15N$ exhibit consistent directional changes (increase or decrease) during some key Phanerozoic transition events, such as the Permian-Triassic boundary (Knies et al., 2013; Du et al., 2021, 2023) and the Late Cretaceous (Meyers et al., 2009; Junium et al., 2018; Du et al., 2025b)."

• Line 400: Quaternary sites also provide extensive latitudinal coverage.

 **Response:** Revised as follows:

"When modern coordinates are converted to paleolatitudes and mapped onto paleogeographic reconstructions, the Cenozoic Era provides the most extensive latitudinal coverage, with the Quaternary period contributing the highest number of sites, followed by the Cretaceous (Figs. 4 and 9)."

• Line 457: After the phrase "we have also provided a software tool on Zenodo", please add "see 8. Code availability".

 **Response:** Added.